# Long Prime–Boost Interval and Heightened Anti-GD2 Antibody Response to Carbohydrate Cancer Vaccine

**DOI:** 10.3390/vaccines12060587

**Published:** 2024-05-28

**Authors:** Irene Y. Cheung, Audrey Mauguen, Shakeel Modak, Ellen M. Basu, Yi Feng, Brian H. Kushner, Nai Kong Cheung

**Affiliations:** 1Departments of Pediatrics, Memorial Sloan Kettering Cancer Center, 1275 York Avenue, New York, NY 10065, USA; modaks@mskcc.org (S.M.); basue@mskcc.org (E.M.B.); fengy@mskcc.org (Y.F.); kushnerb@mskcc.org (B.H.K.); cheungn@mskcc.org (N.K.C.); 2Biostatistics and Epidemiology, Memorial Sloan Kettering Cancer Center, 1275 York Avenue, New York, NY 10065, USA; mauguena@mskcc.org

**Keywords:** ganglioside GD2/GD3 carbohydrate vaccine, high-risk neuroblastoma, anti-GD2 IgG1 titer, beta-glucan, dectin-1 SNP rs3901533

## Abstract

The carbohydrate ganglioside GD2/GD3 cancer vaccine adjuvanted by β-glucan stimulates anti-GD2 IgG1 antibodies that strongly correlate with improved progression-free survival (PFS) and overall survival (OS) among patients with high-risk neuroblastoma. Thirty-two patients who relapsed on the vaccine (first enrollment) were re-treated on the same vaccine protocol (re-enrollment). Titers during the first enrollment peaked by week 32 at 751 ± 270 ng/mL, which plateaued despite vaccine boosts at 1.2–4.5 month intervals. After a median wash-out interval of 16.1 months from the last vaccine dose during the first enrollment to the first vaccine dose during re-enrollment, the anti-GD2 IgG1 antibody rose to a peak of 4066 ± 813 ng/mL by week 3 following re-enrollment (*p* < 0.0001 by the Wilcoxon matched-pairs signed-rank test). Yet, these peaks dropped sharply and continually despite repeated boosts at 1.2–4.5 month intervals, before leveling off by week 20 to the first enrollment peak levels. Despite higher antibody titers, patients experienced no pain or neuropathic side effects, which were typically associated with immunotherapy using monoclonal anti-GD2 antibodies. By the Kaplan–Meier method, PFS was estimated to be 51%, and OS was 81%. The association between IgG1 titer during re-enrollment and β-glucan receptor dectin-1 SNP rs3901533 was significant (*p* = 0.01). A longer prime–boost interval could significantly improve antibody responses in patients treated with ganglioside conjugate cancer vaccines.

## 1. Introduction

The optimal timing between priming and boosting for clinical vaccines is not well understood [1]. For childhood vaccines, the relationship between the prime–boost interval and vaccine efficacy is controversial [2]. Longer mRNA COVID vaccine dosing intervals were associated with improved immunogenicity [3]. In influenza, vaccine effectiveness appeared to decrease with increasing numbers of prior annual vaccinations in some but not all studies [4,5,6]. During influenza seasons, repeat vaccination recipients tended to have lower B cell responses and antibody titers than those receiving primary immunization in their current season only [7,8]. When the induced antibodies were examined, repeat vaccinations were associated with reduced antibody-affinity maturation [9]. Ferrets, an accepted model for human influenza, when given an influenza vaccine boost 10 months after the first immunization, showed less protection than those receiving just the primary immunization [10]. Possible mechanisms include antigenic distance between vaccine and challenge, the original antigenic sin hypothesis, antibody sequestration, and infection block [10]. Because of the genetic drift of viruses and the many limitations of retrospective studies based on health registry data, a large knowledge gap persists regarding the immunologic mechanisms for vaccine interference by inappropriate timing of subsequent boosts.

IgG antibodies against carbohydrates can protect humans against a variety of pathogenic bacteria, including Streptococcus pneumoniae, Hemophilus influenzae B, and Neisseria meningitidis, and their conjugate vaccines have major world-wide health impacts [11,12,13]. While IgG antibodies against tumor associated antigens are proven therapies in cancer, carbohydrate conjugate cancer vaccines have had only measured success [14]. For decades, synthetic ganglioside cancer vaccines have been investigated as therapies for a variety of human cancers [15,16]. The more recent proof of clinical benefits of anti-GD2 IgG antibodies dinutuximab and naxitamab [17,18], together with their FDA approval, provided a strong rationale for the vaccine concept. When conjugated to the highly immunogenic protein scaffold keyhole limpet hemocyanin (KLH), gangliosides GM2, GD2, and GD3 could stimulate antibody responses in patients, typically following a vaccination schedule of repeat vaccine boosts totaling six injections over a period of 6 months. A consistent observation was the rapid leveling off or decrease in antibody titer after each boost, never going beyond the peak level achieved at week 8 in early patient studies. This was true for GM2 [19,20], GD2 [21], GD3 [22], anti-GD3 anti-idiotype [23], GloboH [24,25], and the pentavalent vaccine [26]. The low level and the unexpected decline of antibody titer were ascribed to the low immunogenicity of carbohydrates or the insufficiency of the helper T cell response even in the presence of KLH. Nearly all of these early studies failed to show a definitive proof of benefit in patient outcomes.

In a recent phase II trial, the ganglioside GD2/GD3 vaccine stimulated an antibody response among patients with high-risk metastatic neuroblastoma (HR-NB) who had prior disease progression [27]. In this trial, the addition of oral β-glucan was associated with a 4-fold higher peak titer when compared to prior vaccine trials [21]. In multivariable analyses, a robust anti-GD2 IgG1 antibody titer was associated with longer progression-free survival (PFS) and overall survival (OS), suggesting IgG1 was a protective and beneficial antibody. In a subsequent phase II randomized trial, earlier use of β-glucan during the priming phase significantly enhanced antibody titer by another 2-fold [28]. The favorable impact of the IgG response on survival and the kinetics of the IgG1 response when oral β-glucan was added as an adjuvant were notably different from the findings of old studies of ganglioside vaccines.

In this report, we examined the kinetics of the anti-GD2 IgG1 antibody response of 32 vaccine patients who relapsed on the vaccine either during or after their first enrollment in the trial. Upon regaining clinical remission, they re-enrolled in the same vaccine protocol. We hypothesize that the vaccine injection interval between the first enrollment and re-enrollment could influence the protective antibody response. A longer prime–boost interval could significantly improve antibody responses in patients treated with ganglioside conjugate cancer vaccines.

## 2. Patients and Methods

### 2.1. Patient Inclusion Criteria

Patients with high-risk neuroblastoma who were enrolled and treated in a completed phase II GD2/GD3 vaccine trial at Memorial Sloan Kettering Cancer Center (Clinicaltrials.gov NCT00911560) had the following patient inclusion criteria: They were defined by risk-related treatment guidelines and the International Neuroblastoma Staging System [29,30]. Patients must be in clinical remission in accordance with the International Neuroblastoma Response Criteria (INRC) [29,30] with no evaluable disease by radiographic and histologic assessments to be eligible for enrollment. Institutional Review Board-approved informed consents were obtained from the patients or their guardians.

### 2.2. Study Design

All 32 patients in this analysis had received the GD2/GD3 vaccine. Upon relapse, either on the vaccine or after the completion of the vaccine study, they re-enrolled in the same protocol after achieving clinical remission. In this open-label, single institutional phase I/II study, in combination with oral β-glucan, the GD2/GD3 vaccine conjugated to the carrier protein keyhole limpet hemocyanin (KLH) was mixed with the immunological saponin adjuvant OPT-821 per injection.

The vaccine treatment schema illustrated in Figure 1 identifies the prime and boost intervals. Priming consisted of injections 1, 2, and 3, followed by boosting at injection 4 (week 8), injection 5 (week 20), injection 6 (week 32), and injection 7 (week 52). Each vaccine injection was subcutaneous, with 30 µg GD2 plus 30 µg GD3 conjugated to KLH, mixed with 150 µg/m^2^ OPT-821 [27]. β-glucan (adjuvant) at 40 mg/kg/day was given orally starting from week 6, at a 2-week on and 2-week off schedule until the end of treatment or disease progression.

### 2.3. Blood Collection, Serum Anti-Vaccine Antibody by ELISA, and Dectin-1 SNP Genotyping

Patients had blood drawn before treatment and at weeks 3, 8, 20, 32, and 52 right before vaccine injection. Sera were collected and frozen at −20 °C for batch analyses, and anti-GD2 IgG1 and anti-KLH IgG1 were quantified by ELISA, as detailed in the previous report [27]. The genotyping of dectin-1 (CLEC7A) single nucleotide polymorphism (SNP) rs3901533 was detailed previously [27]. A/A, A/C, and C/C are the 3 genotypes for dectin-1 SNP rs3901533.

### 2.4. Statistical Analysis

Wilcoxon match-pairs signed-rank tests were used to compare the titers of individual patients at various time points during the first enrollment versus re-enrollment. Kruskal–Wallis tests were used for the association between dectin-1 SNP and antibody titer from week 8 after the initiation of β-glucan to the end of therapy or disease progression (PD). This Kruskal–Wallis test is a common non-parametric test (based on rank rather than measured values, equivalent to a one-way ANOVA on ranks) to compare the distribution of a continuous variable between 3 or more groups [31]. Survival rates using time from the first injection of the re-enrollment vaccine through PD or death (PFS) or through death only (OS) were estimated using the Kaplan–Meier method and compared using the log-rank test. The antibody titer was dichotomized using the third quartile cut-point to study its impact on survival as previously described [27]. Patients alive without events were censored on the date of the last follow-up. Median follow-up was calculated using reverse Kaplan–Meier [32].

## 3. Results

### 3.1. Patient Characteristics

Patient characteristics were detailed in Table 1, including gender, age at diagnosis, stage, and disease status at the first enrollment. Also included were the time to progressive disease (PD) on or after the first vaccine trial, the interval between the last treatment of the first enrollment and the first treatment of re-enrollment, as well as the dectin-1 SNP rs3901533 genotypes.

### 3.2. Anti-Vaccine Antibody Response

Patients’ anti-GD2 IgG1 titers were quantified according to schedule during their first vaccine enrollment as well as re-enrollment on the same schedule after re-achieving clinical remission. As shown in Figure 2, antibody titers were markedly different at week 1 (*p* = 0.001), week 3 (*p* < 0.0001), week 8 (*p* < 0.0001), and week 20 (*p* = 0.002) for these 32 relapsed vaccine patients. It was notable that antibody titers during re-enrollment showed a dramatic rise from a baseline of 312 ± 57 ng/mL to a peak of 4066 ± 813 ng/mL by week 3, which dropped sharply and continually before leveling off by week 20, ending at 839 ± 294 ng/mL by week 52. However, the surge was only transient. By week 32 and week 52, the antibody titer was no longer statistically different among these patients between their first enrollment and re-enrollment (*p* = 0.34 and *p* = 0.91, respectively).

Serial titers during the first vaccine enrollment followed by re-enrollment are illustrated in three representative patients (Figure 3). The boost interval averaged 2.8 months (range 1.2–4.5 months). All 7 vaccine injections during re-enrollment were defined as vaccine boosts because priming occurred during the first vaccine enrollment from injection #1 to #3 (Figure 1). The patterns of antibody titers among these patients were strikingly similar, even though patient 3 never completed his first vaccination past week 20 because of relapse. For these 32 relapsed patients, the median interval from the last vaccine dose during their first enrollment to their first vaccine dose after re-enrollment was 16.1 m (range 7.4 to 59.5 m).

As shown in Figure 4, patients with a last vaccine-to-re-enrollment interval of up to 24 months had a higher fold increase in their antibody titer from their last treatment during the first enrollment to the peak titer during re-enrollment when compared with patients whose interval was from 25 months to 60 months. However, given patient heterogeneity and the small sample size, the finding was only suggestive of a possible optimal prime–boost interval.

Besides the anti-hapten (GD2) antibody response, the anti-carrier (KLH) antibody response was also analyzed (Figure 5). It was observed that the anti-KLH antibody increased during the first enrollment, but with re-enrollment, the titer had a marked surge by vaccine injection #3. This high anti-KLH titer persisted through the end of vaccine treatment at week 52, significantly above the week 52 titer during the first enrollment.

### 3.3. Antibody Response and Dectin-1 SNP rs3901533

The association between antibody titers in the presence of oral β-glucan, which was administered to the patients until the end of treatment, and dectin-1 SNP rs3901533 was explored among these relapsed patients. β-glucan was known to enhance antibody-dependent cell-mediated cytotoxicity in the presence of anti-tumor antibodies. Its potential as a vaccine adjuvant is novel. β-glucan binds to the dectin-1 receptor on dendritic cells. Our previous reports found that the anti-GD2 IgG1 response in the vaccine trial correlated with dectin-1 SNP rs3901533 [27,28]. As shown in Figure 6, patients with genotype C/C had a significantly lower average antibody titer than those carrying either the A/A or A/C genotypes (*p* = 0.01). At the first enrollment, a similar trend was observed. However, the association was not statistically significant (*p* = 0.14).

### 3.4. Patient Outcomes

PFS was 51% (95% confidence interval [CI] 36% to 73%) and OS was 81% (CI 66% to 100%) at 54 months (Figure 7). The median follow-up was 26.5 m (range 1.6 to 75.3 m). Among these 32 patients, 15 progressed quickly, with half of the relapses occurring within 5.3 months from the start of their second vaccine trial. Among these 32 relapsed patients, anti-GD2 IgG1 by week 8 was dichotomized using the third quartile cut point as high titer. Using log rank comparison, patients with a high antibody response had a probability of a 30% long-term survival advantage, although this was not statistically significant (*p* = 0.19).

## 4. Discussion

Patients re-enrolled on a GD2/GD3 cancer vaccine after a median wash-out interval of 16.1 months mounted a robust anti-GD2 IgG1 response >10 fold higher than achieved during the first enrollment, only to rapidly decrease to baseline levels with repeated vaccine boosts at 1.2 to 4.5-month intervals. This sharp rise and fall of the anti-GD2 response was accompanied by a surge in the anti-KLH (carrier) titer, which remained high throughout re-enrollment. It was also notable that despite these high anti-GD2 titers during re-enrollment, patients experienced no pain or neuropathic side effects, which are key toxicities encountered during anti-GD2 monoclonal antibody therapy. Understanding the mechanism for this unexpected detrimental effect of short prime–boost intervals on the antibody response to ganglioside vaccines is critical.

Repeat immunizations have been adopted in most vaccine strategies to induce effective life-long protection. A rare exception occurred when one priming dose was sufficient without a boost, such as the yellow fever 17D virus strain (YF17D) and the smallpox vaccinia virus (VACV). Innate cells, particularly those of the myeloid lineage, sense and respond differently to the first priming dose and the subsequent boost [33]. The presence of induced antibodies and memory T cells provides a distinct environment for innate cells at the time of boost, whereas innate cells themselves can exert enhanced antigen-presenting functions long after initial stimulation, which is referred to as trained immunity [33,34,35].

Our data showed that in prior vaccinated patients (i.e., patients who relapsed on the vaccine and re-enrolled to be retreated with the same protocol after achieving clinical remission), a long interval between the first enrollment and re-enrollment was associated with a dramatic surge in the anti-GD2 IgG1 titer. Yet, repeat boosts at short intervals (from 1.2 to 4.5 months) seemed to have a significant negative impact on the anti-GD2 antibody response. During re-enrollment, the anti-GD2 titer dropped before the vaccine boost at injection #4 by nearly 4-fold. Eventually the titer leveled off when the re-enrollment titer and the first enrollment titer became comparable from week 32 through week 52 (Figure 2). These data suggest that while repeat vaccination injections at short time intervals during the priming phase (first 3 weeks) may be effective in establishing memory, once primed, repeat boosting at intervals of 1.2 to 4.5 months did not improve the titer significantly during the first enrollment and in fact may be detrimental by substantially decreasing titers during re-enrollment.

If the mechanism of the damping effect of repeat boosts is understood, new ganglioside cancer vaccine strategies could be tested. One possible explanation for this damping is the neutralizing or blocking potential of anti-GD2 antibodies. Yet, high anti-GD2 or anti-KLH titers did not correlate with damping of the antibody response during either the first enrollment or re-enrollment. Another possible explanation is the exhaustion of B cells or plasma cells with repeat boost, especially if they are scarce (e.g., carbohydrate-specific clones), and in young children after chemotherapy, but that would not explain the strong anti-GD2 surge during re-enrollment. A more likely explanation is a reset of CD4(+) helper T cells when the system became hyperimmune to the protein carrier, thereby turning off the cooperation they normally would provide to hapten-specific B cells. This blunting of the anti-GD2 antibody response is consistent with a feedback inhibition called carrier-induced epitope suppression (CIES) when the vaccine is supercharged by the carrier protein [36]. For our vaccine, it would be KLH, which is used in the current GD2/GD3 vaccine. As evident in Figure 5, anti-KLH antibodies among these patients did surge significantly at the peak of the anti-GD2 IgG1 response during re-enrollment and remained significantly higher than ever achieved during the first enrollment. It is possible that changing the carrier protein to nontoxic diphtheria toxin CRM-197, which is less CIES-prone [37], or alternating CRM-197 with KLH may reduce or eliminate this inhibition—a concept of heterologous prime–boost strategy adopted in other vaccines [38,39,40,41]. It is notable that CRM-197 is a highly effective carrier for conjugate vaccines that has already received FDA approval for a number of vaccines, including Haemophilus influenzae type b (HbOC), Streptococcus pneumoniae (Prevnar 20, Prevnar 15, Prevnar13), and Neisseria meningitidis (Menveo) [42,43,44]. For cancer vaccines, CRM-197 is actively being tested in preclinical testing and clinical trials [45,46]. While KLH is a legacy carrier with production issues, CRM-197 is a recombinant protein commercially available with proven safety [47].

Besides the choice of carrier to evade CIES, the spacing of vaccine boosts is known to influence vaccine response [1,2]. While the chemical nature of the target antigen and adjuvants does matter, schedules that have longer intervals between vaccine doses can lead to more primed and antibody-secreting B cells [48]. With COVID-19 mRNA vaccines, a prime-boost interval of 6 months provided a >5-fold higher titer than the standard one-month interval [3]. With the SARS-CoV-2 mRNA vaccine, a prime–boost interval of 16 weeks boosted mature vaccine-specific B cells when compared to a 4-week interval [49]. In children, antibody responses to acellular pertussis were significantly higher after a 2-month to 4- to 6-month schedule than after an accelerated 2-month to 3- to 4-month schedule [50]. Similarly, the antibody responses to diphtheria–tetanus–pertussis (DTP) vaccination were higher after a 3-month to 5- to 9-month schedule than after an accelerated 2-month to 3- to 4-month schedule [51]. In infants receiving two doses of measles, mumps, rubella, and varicella virus vaccination (MMRV), antibody titers were higher for all components of the vaccine when the doses were given 12 months apart instead of 4 weeks apart [52]. This prime–boost spacing could avoid supercharging by the carrier responsible for CIES.

Besides the carrier and the prime–boost intervals, antibody response to a vaccine can be influenced by host genetics, sex, age at time of vaccination, comorbidities, as well as other vaccine-specific variables including vaccine composition, choice of adjuvants, and vaccination schedule [2]. For neuroblastoma, the carbohydrate antigens, gangliosides GD2 and GD3, were chosen as tumor targets because they are widely expressed among these tumors. They are typically homogeneous and rarely lost, attenuated but recoverable following anti-GD2-directed therapy [53,54]. KLH conjugated to GD2 and GD3 was selected as the vaccine platform because of its proven immunogenicity [55,56], especially when given together with the subcutaneous adjuvant OPT-821, an analog of QS21 [19,20,21,22,23,24,25]. In contrast to previous ganglioside vaccines, oral yeast β-glucan was introduced to exploit the dectin-1 receptor pathway once its safety was proven in a phase I trial [57]. This special gel form of yeast β-glucan was utilized in both the phase I [58] and phase II ganglioside vaccine trials [27,28]. The association of antibody titers with dectin-1 SNP pointed to the adjuvant effect of oral β-glucan on the macrophage/dendritic cell [27,57]. Following oral administration to patients, β-glucan was measurable in the blood [57]. Once in the blood, they were taken up by myeloid cells, as shown in previous preclinical studies [59]. β-glucan was then transported by macrophages to the spleen, lymph nodes, and bone marrow. This systemic activation of innate cells by β-glucan is the basis for the trained immunity model for the GD2/GD3 vaccine strategy [33,34,35], but it also adds another dimension of complexity to the vaccine design.

While understanding the mechanism of antibody response should help refine future vaccine strategies, a relevant question is how this titer after a long prime–boost interval has impacted patient survival. In our previous report of 102 high risk neuroblastoma patients in their first enrollment on the vaccine, there was a clear positive impact of the anti-GD2 antibody response on survival [27]. Whether a long prime–boost interval will influence survival cannot be properly addressed without a randomized trial. However, we should note that relapsed neuroblastoma is not generally survivable even with multi-agent salvage combined chemotherapy/radiation therapy/immunotherapies. In this study, we studied seroconversion after a long prime–boost interval, not expecting these patients to remain in remission, especially since these titer surges rapidly dropped back to their first enrollment levels. Nevertheless, at 54 months, PFS was 51% and OS was 81%. One can only speculate whether the >10-fold improvements in titer, though transient, were enough to prolong remission and survival in a subset of these patients. In the current study, comparisons using the log rank test did show a better overall survival among those with a higher antibody titer, although statistically insignificant, likely because of patient heterogeneity and a small patient number.

Much work remains to be done to elucidate the underlying mechanism of seroconversion to carbohydrate cancer vaccines, including adjuvants, delivery systems, schedules, and the role of the microbiome [60,61,62,63]. For ganglioside conjugate vaccines, prime–boost interval deserved closer attention as a variable well known in the vaccine field [64]. Mathematical models, including artificial intelligence, to explore the optimal dose, adjuvant, delivery system, or boost intervals used assumptions that required validation [65]. Empirically, optimizing the prime–boost schedule seems necessary if higher and more persistent titers are the vaccine goals. Previous ganglioside vaccines that have failed clinically deserve to be revisited with more attention paid to innate immunity adjuvants and prime–boost intervals.

## Figures and Tables

**Figure 1 vaccines-12-00587-f001:**
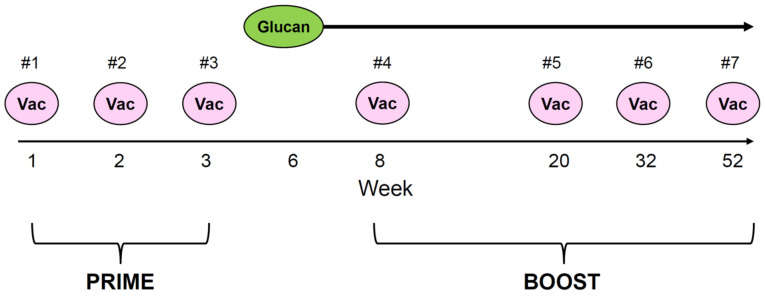
This was an open-label, single institutional phase I/II study of the GD2/GD3 vaccine in combination with oral β-glucan for high-risk neuroblastoma (clinicaltrials.gov NCT00911560). The vaccine treatment schema identified the prime and boost intervals, with a total of 7 injections spanning 52 weeks. Each subcutaneous vaccine injection consisted of 30 µg GD2 plus 30 µg GD3 conjugated to carrier protein KLH (keyhole limpet hemocyanin), mixed with 150 µg/m^2^ OPT-821 (immunological saponin adjuvant). β-glucan (adjuvant) at 40 mg/kg/day was given orally starting from week 6, at a 2-week on and 2-week off schedule until the end of treatment or disease progression.

**Figure 2 vaccines-12-00587-f002:**
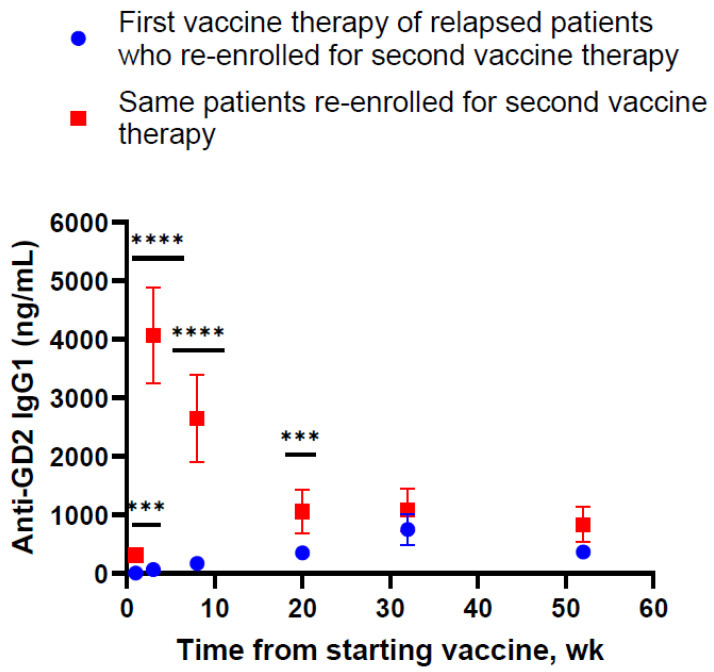
Thirty-two patients who were treated with the vaccine had relapsed. Upon achieving clinical remission, they were re-enrolled in the same vaccine protocol. Their anti-GD2 IgG1 titer was measured by ELISA and expressed as mean ± SEM. Using the Wilcoxon matched-pairs signed-rank test, a statistically significant difference was observed at week 1, week 3, week 8, and week 20 for these patients by comparing their antibody titers during their first vaccine therapy and their second vaccine therapy. *** *p* ≤ 0.002, **** *p* ≤ 0.0001.

**Figure 3 vaccines-12-00587-f003:**
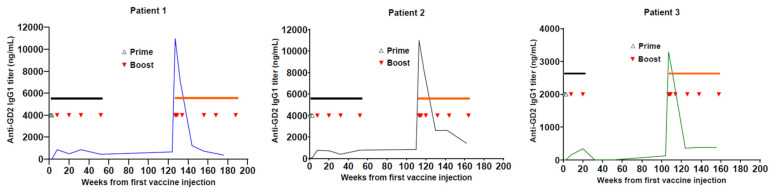
Illustration of three representative patients who relapsed on first vaccine therapy, and they re-enrolled in the same vaccine therapy after achieving clinical remission. The anti-GD2 IgG1 titer was quantified for patients #1, #2, and #3 from their first injection at the first enrollment (

) to the last injection at re-enrollment (

). The timing of prime and boost was indicated. Months from the last upfront vaccine to the first injection during re-enrollment were as follows: 16.6 m for patient #1, 13.4 m for patient #2, and 12.0 m for patient #3.

**Figure 4 vaccines-12-00587-f004:**
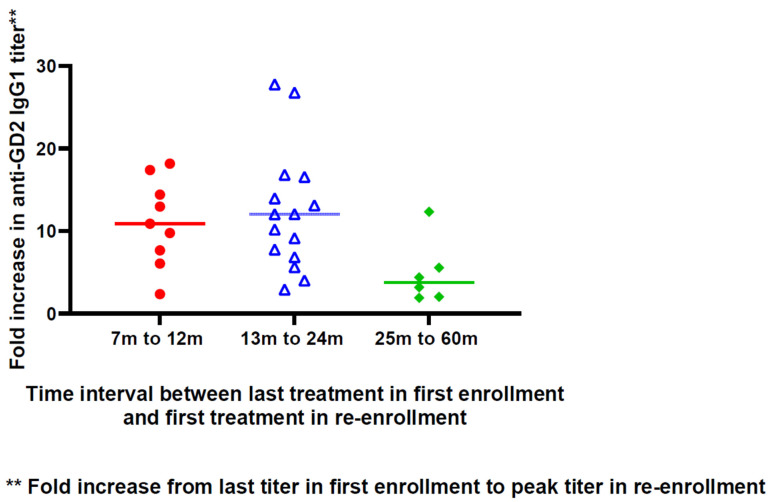
Relationship between the interval in months from the last upfront vaccine to the first injection during re-enrollment and the fold increase from the last titer in the first enrollment to the peak titer in the re-enrollment for individual patients. Two patients with no detectable antibody titer throughout their two vaccine treatments were excluded from this analysis.

**Figure 5 vaccines-12-00587-f005:**
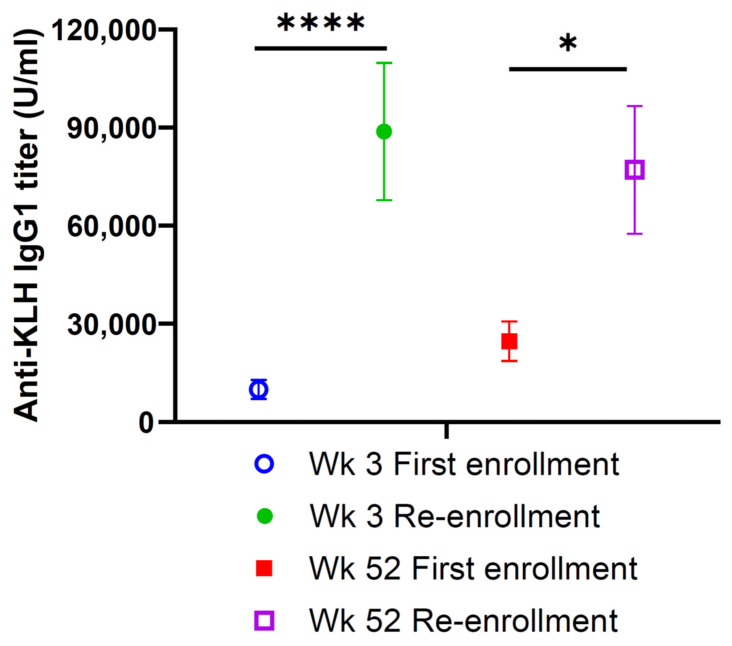
Anti-KLH IgG1 titer among patients who relapsed on vaccine and re-enrolled in the same vaccine therapy after achieving clinical remission. Their anti-KLH IgG1 titer was measured by ELISA and expressed as a mean ± SEM. Using the Wilcoxon matched-pairs signed-rank test, a statistically significant difference was observed when comparisons were made between different time points. * *p* = 0.04, **** *p* ≤ 0.0001.

**Figure 6 vaccines-12-00587-f006:**
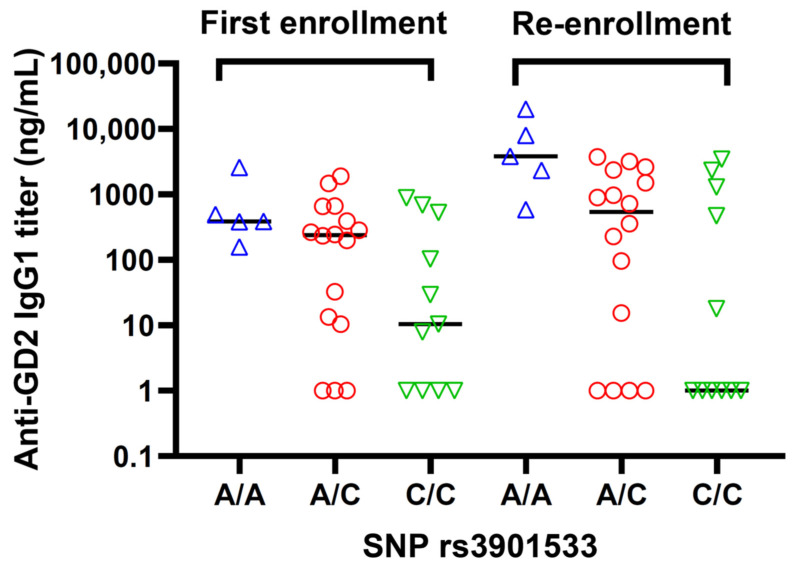
Association between antibody titer and SNP rs3901533 of dectin-1 (CLEC7A) polymorphism among 32 relapsed patients. The average antibody titer from week 8 after the initiation of β-glucan to the end of vaccine treatment or disease progression was tabulated for individual patients. Statistical analyses were performed using the Kruskal–Wallis test. First enrollment (*p* = 0.14); re-enrollment (*p* = 0.01).

**Figure 7 vaccines-12-00587-f007:**
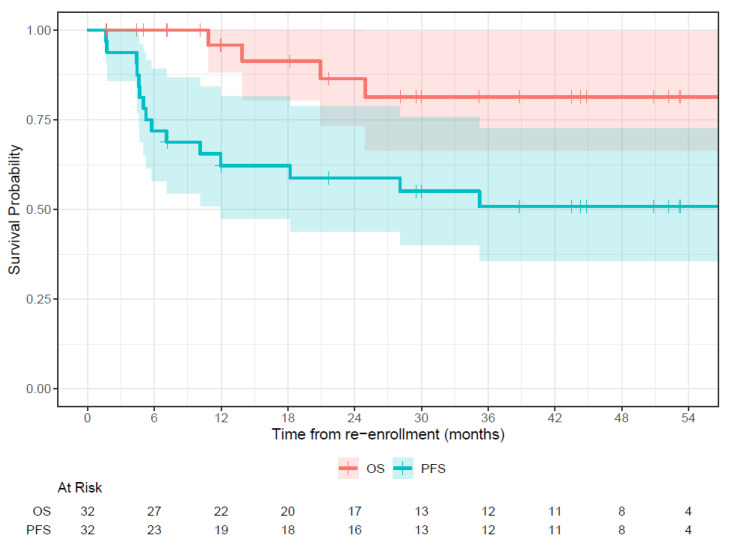
Survival outcome after second vaccine therapy among 32 relapsed vaccine patients; progression-free survival (PFS) and overall survival (OS).

**Table 1 vaccines-12-00587-t001:** Characteristics of 32 high-risk neuroblastoma patients who relapsed on/after vaccine treatment, achieved clinical remission, and re-enrolled in the same vaccine regimen.

Characteristics	N = 32 ^1^
Sex	
Female	13 (41%)
Male	19 (59%)
Diagnosis	2010-09-24 to 2018-05-02
Age at diagnosis (y)	3.3 [0.1–10.4]
<18 months	4 (12%)
≥18 months	28 (88%)
Neuroblastoma stage	
3	1
4	31
Disease status at the first vaccine trial	
First CR	16 (50%)
Second CR with prior PD	16 (50%)
Time to PD on/after the first vaccine trial (m)	10.7 [2.3–59.1]
<12	19 (60%)
12–30	10 (31%)
>30	3 (9%)
Dectin-1 SNP rs3901533	
A/A	5 (16%)
A/C	16 (50%)
C/C	11(34%)
First vaccine date after the first enrollment	2014-06-04 to 2019-09-17
First vaccine relapse date	2015-10-05 to 2020-05-11
First vaccine date after re-enrollment	2017-01-12 to 2021-07-13
Interval between the last vaccine dose of the first enrollment and the first vaccine dose of re-enrollment (m)	16.1 [7.4–59.5]
^1^ N (%); Median [Range]	
CR, clinical remission; PD, progressive disease	
SNP, single nucleotide polymorphism	

## Data Availability

The datasets used and/or analyzed during the current study include patient identifying information. Any data not provided in this manuscript and produced for this research is available in deidentified format from the corresponding author on reasonable request.

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
