# Peer review of "Long Prime–Boost Interval and Heightened Anti-GD2 Antibody Response to Carbohydrate Cancer Vaccine"

_vaccines, 2024, doi:10.3390/vaccines12060587_

Round 1

Reviewer 1 Report

Comments and Suggestions for Authors  

The manuscript corresponds to a brief report on the level of the IgG response induced against carbohydrate antigen GD2 all along a vaccination protocol based on 7 injections (3 primes at one-week
interval each + 4 boosts). The cohort comprises 32 children who relapsed on neuroblastoma after a first, identical vaccination protocol. The reported data fit with the scope of the journal but remain limited. The authors initially observe a high increase of the anti-CD2 IgG response (in agreement with what is usually observed during a booster effect). However, IgG titers rapidly dropped to reach the
baseline level. Since the median interval between last vaccine dose of first enrollment and first vaccine dose of re-enrollment is 16 months, the authors suggest that increasing the prime-boost interval might
contribute to improve the Ab response for cancer vaccines.

1) As emphasized by the authors, the immunization protocol influences the effectiveness of a vaccine without there being a clear consensus. They cite examples which refer to protein sub-unit vaccines
and mainly to influenza vaccine.
Is it possible to add data dealing with glycoconjugate vaccines (pneumococcal, meningococcal) more in line with the herein tested GD2-KLH glycoconjugate vaccine?
2) That a boost is observed after the first vaccine dose of re-enrollment suggest that the memory response is in place but that some still unknown immune mechanisms abrogate or, at least, mitigate the response. The authors list several hypotheses some of these relating to possible interference from the carrier. Since the authors have all the sera in hands, I think it should be an easy experiment to determine the anti-carrier protein IgG response. Providing such data would be useful to support or eliminate such hypotheses in the discussion section.

Typos

Figure 1: What does the “23” on the x-axis mean?

Figure 5: Vaccine (instead of vac-cine)

Author Response

Please review attached pdf.  Thanks.

Reviewer 2 Report

Comments and Suggestions for Authors

Carbohydrate ganglioside GD2/GD3 cancer vaccine adjuvanted by β-glucan stimulates anti-GD2-IgG1 antibodies that correlates with improved progression-free survival (PFS) and overall survival (OS) among patients with high-risk neuroblastoma. The authors report thirty-two patients who relapsed on vaccine (first enrollment) and were re-treated on the same vaccine protocol (re-enrollment). Titers during first enrollment peaked by week 32 to 751±270 ng/mL which plateaued despite vaccine boosts at 1.2-4.5 months intervals. After a median wash-out interval of 16.1 months from the last vaccine dose during first enrollment to first vaccine dose during re enrollment, anti-GD2-IgG1 antibody rose to a peak of 4,066±813 ng/mL by week 3 following re-enrollment (P<0.0001). Yet, these peaks dropped sharply and continually despite repeated boosts at 1.2-4.5 months intervals, before leveling off by week 20 to first enrollment peak levels. Despite higher antibody titers, patients experienced no pain or neuropathic side effects, which were typically associated with immunotherapy using monoclonal anti-GD2 antibodies.
PFS was estimated to be 51%, and OS was 81%.

The association between IgG1 titer during re-enrollment with β-glucan receptor dectin-1 SNP rs3901533 was significant.

A longer prime-boost interval could significantly improve antibody response in patients treated with ganglioside conjugate cancer vaccines.

Interesting paper by a team widely involved in immunotherapy in high risk neuroblastoma. Interesting PFS and OS for these bad prognosis patients. Need to be confirmed at a large scale.

Author Response

Please review the attached pdf.

Reviewer 3 Report

Comments and Suggestions for Authors

The manuscript from I. Y. Cheung et al. aims to investigate the kinetics of anti-GD2 IgG1 antibody response in 32 neuroblastoma patients who relapsed on their first vaccine treatment. Their work focused on the link between antibody titer and 1) prime/boost intervals as well as 2) interval between first enrollment and re-enrollment, for an improved progression-free survival and overall survival of the patients. 

A major revision should be addressed before publication:

- Introduction: since the manuscript is focused on cancer vaccines, authors should add more data on antibody response in this field.

- The sentence at the end of the introduction summarizes the hypothesis of the manuscript but not the main findings. Please add more details.

- Although in the results section, it is clearly stated that “relapsed patients have been re-enrolled on the same schedule of the first vaccine”, this is not reported in the patients and methods section. Please describe it accordingly, and possibly implement it also in Figure 1;

- Please better describe in the patients and methods section, the statistical tests used for data comparison and association. Especially the “Kruskal-Wallis rank sum test” needs much more explanation even in the presentation of the results (A/A, A/C, C/C genotypes are not described). Statistical significance among the different genotypes during the first- and re-enrollment is also not clear;

- Please shortly describe what “achieving clinical remission” means, to better understand the evaluation and treatment intervals;

- Figure 5 and the survival outcome of patients after the second vaccine therapy is barely discussed in the results part; this is crucial since I assume this is linked to the sentence in the introduction “We hypothesize that the vaccine injection interval between first enrollment and re-enrollment could influence protective antibody response”;

- Please reformat the Figure 3’ legend; Moreover, could you elaborate on the selection of the 3 patients? Could you comment on the phrase “The patterns of antibody titers among these patients were strikingly similar” related to these 3 patients? Is it possible to plot the data of all 32 patients or at least divide them by different intervals between first enrollment and re-enrollment?;

- Please revise the discussion by first summarizing your results and then discussing them in the specific context;

- The first 28 lines of the discussion part are a repetition of part of the introduction. Either revise them or move them to the introduction;

- The discussion presents the following sentence: “It is notable that CRM-197 is a highly effective carrier for conjugate vaccines that has already received FDA approval”. Please add a reference and shortly describe the type of tumor currently in use;

- The final message in the discussion part “Previous ganglioside vaccines that have failed clinically deserve to be revisited with more attention paid to innate immunity adjuvant and prime-boost intervals.” is pretty clear but needs references. Specifically, which other ganglioside vaccines? Moreover, has this attention to adjuvants and prime-boost intervals been also considered in other tumor vaccines?

Author Response

Please review the attached pdf.

Round 2

Reviewer 1 Report

Comments and Suggestions for Authors

The authors have added some references on glycoconjugate vaccines (referred to as 11 to 14) in response of one of my request. I thank them for this effort. However, although interesting, these references do not address the problem of the vaccination schedule.

Are there no studies discussing the effects on response effectiveness related to the intervals between each injection? If studies on glycoconjugate vaccine schedules exist, it would make sense to feed the manuscript with references and discussion.

Reviewer 3 Report

Comments and Suggestions for Authors

All the points raised in my previous revision were successfully answered by the authors. 

Author Response

RESPONSE:  We thank the reviewer for stating that all the critiques were successfully answered.